# Endocytosis and Digestion in Carnivorous Pitcher Plants of the Family Sarraceniaceae

**DOI:** 10.3390/plants8100367

**Published:** 2019-09-24

**Authors:** Marianne Koller-Peroutka, Stefanie Krammer, Anselm Pavlik, Manfred Edlinger, Ingeborg Lang, Wolfram Adlassnig

**Affiliations:** 1Core Facility Cell Imaging and Ultrastructure Research, University of Vienna, Althanstraße 14, A-1090 Vienna, Austria; stefanie.goergl@univie.ac.at (S.K.); anselm.pavlik@univie.ac.at (A.P.); wolfram.adlassnig@univie.ac.at (W.A.); 2HBLFA for horticulture and the Austrian Bundesgärten Schönbrunn, Schloß Schönbrunn, A-1130 Vienna, Austria; manfred.edlinger@bundesgaerten.at; 3Department of Ecogenomics and Systems Biology, University of Vienna, Althanstraße 14, A-1090 Vienna, Austria; ingeborg.lang@univie.ac.at

**Keywords:** *Sarracenia*, *Heliamphora*, *Darlingtonia*, endocytosis, carnivorous plants, pitcher traps, uptake, digestion, FITC-BSA, methylene-blue

## Abstract

Highly evolved carnivorous plants secrete digestive enzymes for degradation of trapped animals and absorb whole macromolecules from their prey by means of endocytosis. (1) Background: In the pitcher-plant family Sarraceniaceae, the production of enzymes is dubious and no evidence for endocytosis is known so far. (2) Methods: *Heliamphora nutans*, *Darlingtonia californica*, and nine taxa of *Sarracenia* are tested for cuticular pores, and for protease and endocytosis of the fluorescent protein analogue FITC-BSA, after 10–48 h of stimulation. (3) Results: Cuticular pores as a prerequisite for enzyme secretion and nutrient uptake are present in all tested species. Permeable cells form clusters in the inner epidermis of the pitchers, but are only little differentiated from impermeable epidermis cells. Proteases are found in *S. psittacina*, *S. leucophylla*, *S. minor*, *S. oreophila*, *S. alabamensis*, *H. nutans*, *D. californica* lacking only in *S. flava* and in *S.*
*purpurea* ssp. *purpurea*, *S. purpurea* ssp. *venosa*, *S. rosea*, where enzyme production is possibly replaced by degradation *via* the extraordinary diverse inquiline fauna. *S.*
*leucophylla*, *S. minor*, *S. oreophila* exhibit both protease production and endocytosis; *S. psittacina*, *S. alabamensis*, *H. nutans*, *D. californica* produce proteases only; no single species shows endocytosis without protease production. (4) Conclusions: Protease secretion seems to be a prerequisite for endocytotic nutrient uptake. Transport of FITC-BSA absorbed by endocytosis towards the vascular tissue of the trap leaves suggests that endocytosis of nutrients is more than a side effect of enzyme secretion.

## 1. Introduction

The American pitcher plant family Sarraceniaceae comprises three genera of pitcher plants with at least 35 species: *Darlingtonia*, *Heliamphora*, and *Sarracenia. Sarracenia* includes eleven species and a multitude of subspecies, varieties, and hybrids, and is native to eastern and northern North America. *Heliamphora* comprises 23 species, which grow in northern South America, whereas *Darlingtonia* is a monotypic genus that appears only in the northwestern United States [1,2,3]. Within the family of Sarraceniaceae, prey capture is performed by cone shape leaves, which are superficially similar to the pitcher traps of *Nepenthes* and *Cephalotus*, but less sophisticated. This is not only true on the morphological level, but possibly also for the physiology. Since the first studies on *Sarracenia*, the production of digestive enzymes for prey degradation was questioned [4]. Gallie and Chang [5] report nucleases, proteases, and phosphatases in *S. purpurea*, whereas Smith [6] found no evidence for an accelerated degradation of meat in the fluid of *S. purpurea* compared to distilled water. On the other hand, a multitude of mutualistic pitcher inquilines has been described (summarized by [7,8,9]), ranging from bacteria to insect larvae, which contribute to prey degradation. Therefore, most authors assumed that enzyme production played at maximum a minor role in *Sarracenia*. Recent research, however, suggested the presence of digestive enzymes at least in part of *Sarracenia* [10,11].

Furthermore, prey utilization in *Sarracenia* seems to be less sophisticated with regards to the absorption of nutrients into living cells. Uptake is performed *via* selective carriers for inorganic ions and small organic molecules [12], as in presumably all other carnivorous plants [13]. Endocytosis, however, was suggested as a more advanced mechanism for nutrient uptake in carnivorous plants, which would enable the absorption of whole proteins or large fractions thereof [14]. Using fluorescent labelled proteins, the authors of this study were able to confirm this hypothesis by detecting endocytotic nutrient uptake in multiple genera of carnivorous plants, including *Drosera*, *Dionaea*, *Nepenthes*, *Cephalotus*, *Genlisea*, and *Utricularia*, but not in *S. purpurea* [15].

In eukaryotic cells, the secretion of proteins, including digestive enzymes, is usually performed *via* exocytosis [16]. Since the membrane surface of a non-growing cell is constant, exocytosis is usually coupled with endocytosis, resulting in the uptake of the external medium, including any dissolved compounds. In all aforementioned carnivorous plants with endocytotic nutrient uptake, the secretion of digestive enzymes has been demonstrated beyond reasonable doubt (summarized by [11,13]). Thus, the capability to absorb nutrients via endocytosis may have developed primarily as a side effect of the exocytosis of digestive enzymes.

This study aims to test the hypothesis of a relationship between enzyme secretion and endocytotic uptake of whole proteins in the genus *Sarracenia*, supplemented by other Sarraceniaceae. The primary objective is to establish a correlation between autonomous digestion and endocytosis. Secondary objectives include screening for digestion, the fate of the absorbed proteins after endocytosis, and the characterization of pores in the impermeable [17] cuticle of the traps, which are a prerequisite of any nutrient uptake [18,19] regardless of the cellular mechanism. 

## 2. Results

The habitus of seven selected species are depicted in Figure 1.

Methylene blue staining (Figure 2 and Appendix A) demonstrated the presence of cuticular pores in all investigated species. Pores were restricted to the absorptive zone for uptake which is, following the terminology of Juniper et al. [20], zone 4 for *Sarracenia* and *Heliamphora*, and zone 5 for *Darlingtonia* (Appendix A). These cuticular pores are completely lacking in the retention zone of the pitchers. Though no morphologically differentiated glands are developed in the absorptive zones of the pitchers in Sarraceniaceae, three types of epidermal cells can be distinguished: (1) Trichoblasts protruding into retentive hairs (opposed to atrichoblasts) never absorb methylene blue, neither to the cell wall, nor to the cytoplasm. (2) Some epidermal cells do not absorb methylene blue into the cytoplasm, but sometimes into the cell wall. (3) Other epidermal cells of the same shape but of a slightly smaller surface area (Appendix A) accumulate methylene blue in the cytoplasm. The absorbing cells usually form isolated clusters surrounded by non-absorbing cells (Figure 2 and Appendix A). 

No evidence for digestion of gelatin emulsion (of analogue photographic film) was found in *S. purpurea*, *S. rosea*, and *S. flava*. In all other analyzed species of *Sarracenia*, as well as *Heliamphora nutans* and *Darlingtonia californica*, degradation of the gelatin indicated secretion of digestive enzymes upon stimulation (Figure 3c–e). No evidence for absorption of FITC-BSA into the cytoplasm was found in *S. purpurea* (Figure 4a), *S. rosea*, *S. flava*, *S. psittacina*, *S. alabamensis*, and *Heliamphora nutans* and *Darlingtonia californica*. In these species, FITC-fluorescence was visible in the cell walls of normal epidermal cells but not in the trichoblasts. In *S. leucophylla*, *S. minor*, and *S. oreophila* uptake of FITC-BSA was found in isolated clusters of epidermal cells (Figure 4b–d), similar to the uptake of methylene blue. Quantitative analysis of the FITC-fluorescence displayed highly significant differences between absorbing and non-absorbing cells in *S. oreophila*, and compared to *S. purpurea* (*p* < 0.01 in all cases, see Figure 5).

Uptake patterns differed between cells of the same section: FITC-BSA might be found in vesicles of varying size, in the cytoplasm, or in the vacuole. The highest fluorescence intensity was found in the central vacuole, whereas chloroplasts and the nucleus remain dark. At an early stage of uptake, after up to 15 h (Figure 4e), FITC-BSA was exclusively found in the epidermal cells. Later on, the mesophyll, and in particular the vascular bundles, are stained as well (Figure 4f). Table 1 summarizes all results.

## 3. Discussion

In addition to the trapping of animals, carnivorous plants, by definition, digest and utilise their prey [1,20]. The absorption of small molecules into the epidermis by means of methylene blue staining was confirmed for all investigated species, as was to be expected from earlier results [12,21,22]. Protease production and digestion within 48 h after stimulation seems to be present in the majority of Sarraceniaceae, including *Darlingtonia* and *Heliamphora* [23], but lacking in the closely related [24] species *S. rosea* and *S. purpurea*. *S. rosea* is a sister to both *S. purpurea* ssp. *purpurea* and *S. purpurea* ssp. *venosa* [2,24], which show similar physiological features in this study, and are enclosed in the *Sarracenia purpurea* complex [24]. Previous data on protease production in *S. purpurea* are contradictory; one study which found evidence for protein digestion [5] suggests that protease activity, if any, peaks after >4 days. A recent study by Young et al. [25] indicates that protease activity in the pitcher fluid of *S. purpurea* remains at baseline levels if antibiotics are added.

Uptake of whole proteins was restricted to only three *Sarracenia* species (Table 2). All species with endocytosis also secreted proteases, but not vice versa. Secretion of proteases by exocytosis can be expected to be accompanied by the recycling of membranes by endocytosis. However, this membrane recycling alone seems not to result in the uptake of detectable amounts of FITC-BSA. The absorption of visible quantities seems to require specific cellular mechanisms, which are present only in a few species of *Sarracenia*, and are accompanied by mechanisms for the transport of FITC-BSA towards the mesophyll.

On the cellular level uptake into the cytoplasm is restricted to specialized cells. Though these permeable cells are virtually indistinguishable from the surrounding impermeable cells, they seem to be as specialized as the morphologically differentiated specialized glands of other carnivorous plants. Within the cells, the uptake differs from the pattern found in Droseraceae and Nepenthaceae [15]: in these families, a multitude of small vesicles is found which fuse to large endosomes, partially displacing the central vacuole. In *Sarracenia*, fluorescent vesicles seem to be inconspicuous, whereas FITC-BSA accumulates in the cytoplasm and particularly in the central vacuole. Thus, the degradation of the absorbed protein seems to follow a different pathway. However, transport towards the vascular system is found as in *Drosera* or *Genlisea* [15].

## 4. Materials and Methods 

### 4.1. Plant Material

The following Sarraceniaceae were studied (see Figure 1): *Sarracenia purpurea* ssp. *purpurea, Sarracenia purpurea* ssp. *venosa, Sarracenia rosea* (= *Sarracenia purpurea* ssp. *venosa* var. *burkii*), *Sarracenia psittacina*, *Sarracenia flava* var. *flava*, *Sarracenia leucophylla*, *Sarracenia minor* var. *minor*, *Sarracenia oreophila* var. *alba*, *Sarracenia alabamensis* (= *Sarracenia rubra* ssp. *alabamensis*), *Heliamphora nutans*, *Darlingtonia californica*.

All test plants were nursed at the botanical collection of the HBLFA for horticulture and the Österreichischen Bundesgärten, Vienna, Austria. During the experiments, plants were located at the greenhouse of the Biocenter Althanstraße, University of Vienna, Austria.

### 4.2. Methylene Blue-Staining for Detection of Cuticular Pores

Cuticular pores were detected by using a 0.1–1% aqueous solution of methylene blue as described by Adlassnig et al. [15]. 

### 4.3. Test for Proteases

The presence of digestive enzymes (i.e., proteases) was tested with conventional photographic film, as described by Meyers-Rice [26], slightly modified. *Sarracenia* pitchers were isolated from the plant, cut in halves with a razor blade, functional zone was selected, wetted with a 1% solution of the protein BSA (Bovine Serum Albumin—cold ethanol fraction pH 5.2, ≥96%, Lot # SLBWW2269, as well as Bovine Serum Albumin—lyophilized powder, ≥96%, Lot #SLBJ8588V, both SIGMA Life Science) in tap water, in order to stimulate protease secretion. Growth of trap inhabiting bacteria, which might falsify the outcome by producing additional proteases, was inhibited by adding tetracycline (Tetracycline.HCl, research grade, Serva Lot#35866, concentration 0.1 mg per l). Pieces of photographic film ILFORD XPI 400 were assembled with the gelatin emulsion side onto the tissues of the test plants and mostly fixed with staples (see Figure 3). Subsequently, these prepared trap tissues were incubated in a humid chamber for 10–48 h, the film was developed by using a conventional developer solution (Kodak TMax RS developer). Secretion of protease led to digestion of the gelatin emulsion resulting in transparent areas, whereas undigested intact gelatin emulsion remains opaque.

### 4.4. FITC-BSA Staining to Test for Endocytosis and Uptake

FITC-BSA (Albumin, Fluorescein isothiocyanat Conjugate from Bovine, SIGMA Product Number A9771) contains the fluorescent dye FITC (molecular weight 389.4), which is conjugated with the protein BSA (Bovine Serum Albumin, molecular weight of about 66,000). A 1% FITC-BSA solution—diluted with tap water or, if available, with the native filtrated pitcher fluid, was applied. The experimental design was similar to Adlassnig et al. [15]: A droplet FITC-BSA solution was placed on those pitcher tissues where cuticular discontinuities had been found. The samples were incubated at 28–30 °C in humid chambers under dark conditions to prevent bleaching. After 18–48 h, the samples were thoroughly rinsed, and both surface and longitudinal hand sections were made and analyzed with a confocal scanning microscope (CLSM, Leica TCS SP5 DM-6000 CS).

As negative control, a 1% BSA-solution was applied and subsequently treated and analyzed like the fluorescent test series.

Fluorescence of FITC-BSA was excited with the 496 nm laser-line from the argon-laser and detected at 500–550 nm. Fluorescence intensity as well as cell area were quantified by using the Leica LAS AF Lite–Version: 3.3.10134, and compared between different treatments and stainings by using the statistical analysis software STATA 14.2. Since all samples deviated significantly from normal distribution (Skewness–Curtosis test), the non-parametric Kruskal–Wallis test with the post-hoc Dunn test were used to test for differences. *p* < 0.01 was regarded as highly significant. Cohen’s κ was used to assess the correlation between the presence or lack of digestion and endocytosis. The non-parametric Mann–Whitney Test was used to compare the area of permeable and impermeable cells.

## 5. Conclusions

Most members of Sarraceniaceae produce protease, but only a few species of *Sarracenia* are capable of endocytotic nutrient uptake.

Endocytotic nutrient uptake seems to have developed from membrane recycling in the frame of the exocytosis of enzymes.

In Sarraceniaceae, physiologically specialized epidermal cells fulfil the same functions as morphologically differentiated glands in other carnivorous plants.

## Figures and Tables

**Figure 1 plants-08-00367-f001:**
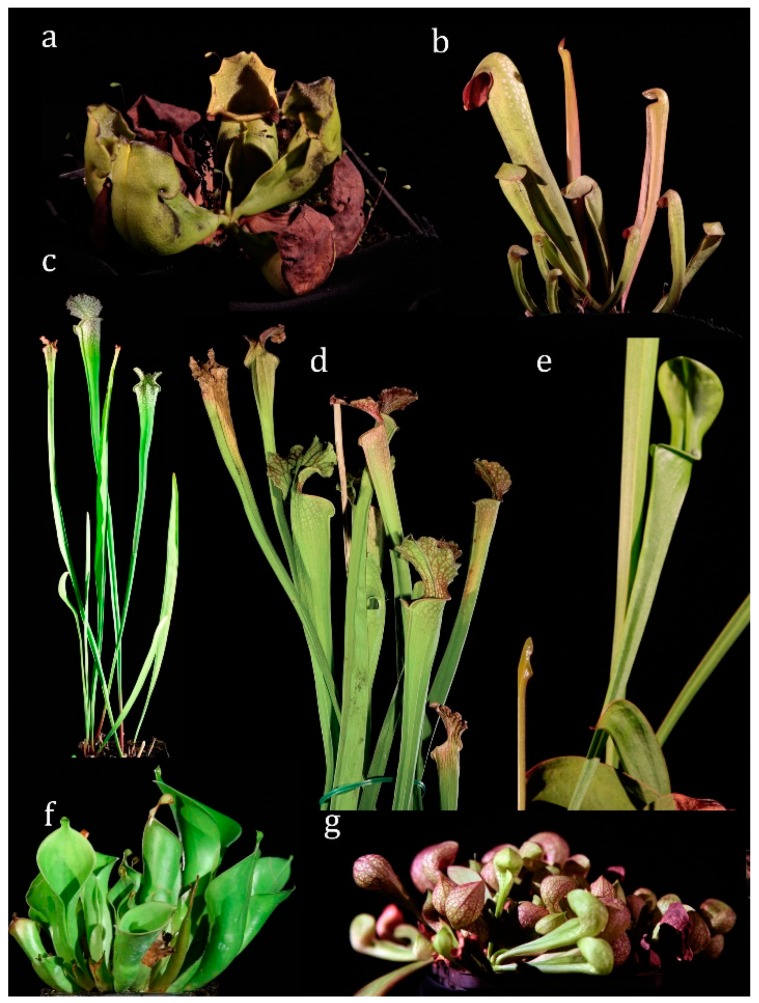
Habitus of tested Sarraceniaceae. (**a**) *Sarracenia purpurea* ssp. *purpurea* (**b**) *Sarracenia minor* (**c**) *Sarracenia leucophylla* (**d**) *Sarracenia flava* (**e**) *Sarracenia oreophila* (**f**) *Heliamphora nutans* (**g**) *Sarracenia psittacina*.

**Figure 2 plants-08-00367-f002:**
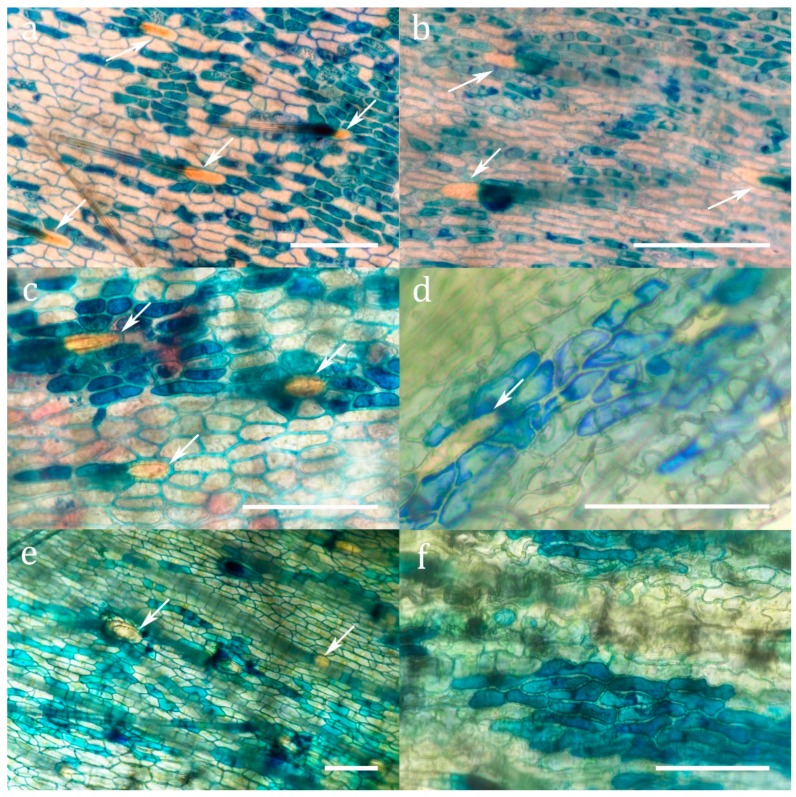
Methylene blue staining of cuticular pores in the absorptive zone (**a**) *Sarracenia alabamensis* (**b**) *Sarracenia leucophylla* (**c**) *Sarracenia minor* (**d**) *Sarracenia oreophila* (**e**) *Darlingtonia californica* (**f**) *Sarracenia psittacina*. Arrows indicate trichoblasts. Scale bar: 100 µm.

**Figure 3 plants-08-00367-f003:**
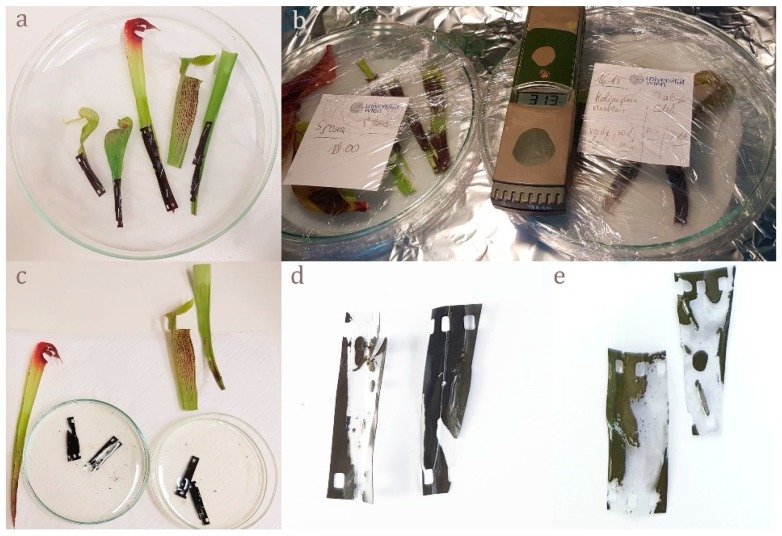
Enzyme test for proteases; (**a**) preparing *Darlingtonia californica*, *Sarracenia psittacina*, *Sarracenia minor*, and *Sarracenia alabamensis* for the enzyme test by attaching little pieces of film wetted with BSA onto the absorptive zones of the pitchers; (**b**) incubation in wet chambers for 18–48 h; (**c**) developing of the film pieces in a film developer; (**d**) positive enzyme test for *S. minor*; (**e**) positive enzyme test for *Darlingtonia californica*.

**Figure 4 plants-08-00367-f004:**
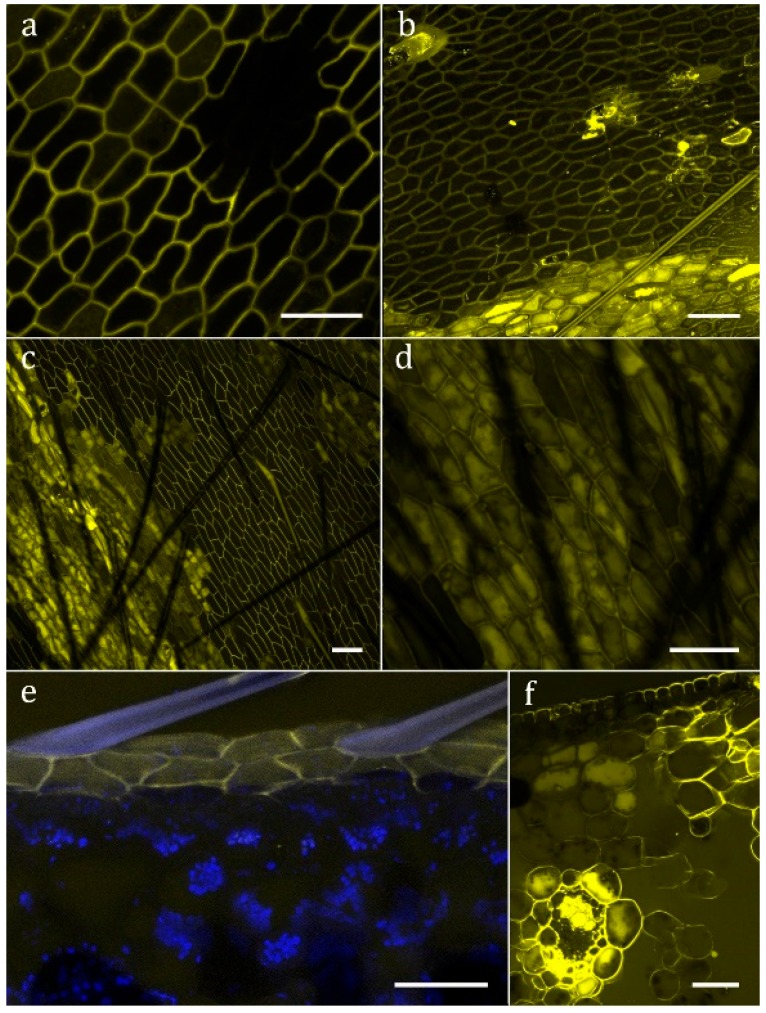
Uptake of the fluorescent protein FITC-BSA. The yellow color shows FITC-fluorescence in the detection wavelength 500–550 nm. (**a**) *Sarracenia purpurea* epidermal cells show no uptake; (**b**) *Sarracenia minor*; (**c**,**d**,**f**) *Sarracenia oreophila*; (**e**) *Sarracenia leucophylla* (the blue color shows autofluorescence of cuticle and chlorophyll, detection wavelength 626–672 nm). Scale bar: 50 µm.

**Figure 5 plants-08-00367-f005:**
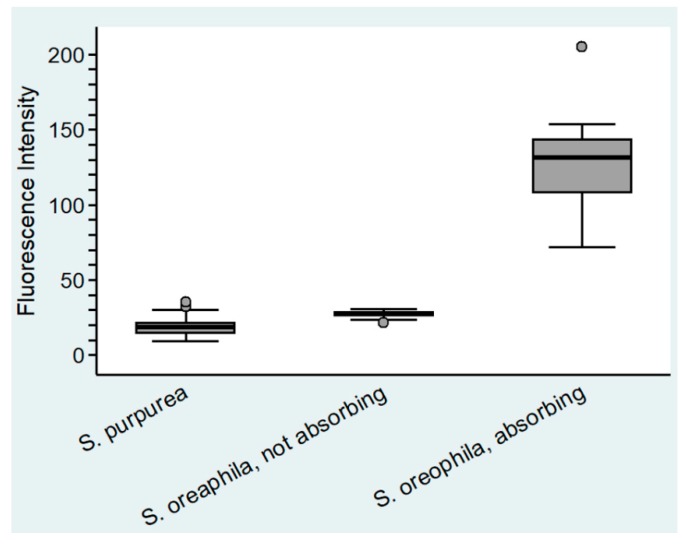
Fluorescent intensity in the cytoplasm of epidermal cell of *Sarracenia purpurea* and *Sarracenia oreophila* (n = 20).

**Table 1 plants-08-00367-t001:** Results of the physiological analysis for enzyme formation, endocytosis, and methylene blue uptake.

Species	Cuticular Pores	Digestive Enzymes	Endocytosis
*S. purpurea* ssp. *purpurea*	+	-	-
*S. purpurea* ssp. *venosa*	+	-	-
*S. rosea*	+	-	-
*S. psittacina*	+	+	-
*S. flava* var. *flava*	+	-	-
*S. leucophylla*	+	+	+
*S. minor* var. *minor*	+	+	+
*S. oreophila*	+	+	+
*S. alabamensis*	+	+	-
*Heliamphora nutans*	+	+	-
*Darlingtonia californica*	+	+	-

**Table 2 plants-08-00367-t002:** Number of species/subspecies showing or lacking digestion or endocytosis respectively.

	Digestion Lacking	Digestion Present	Total
Endocytosis lacking	4	4	8
Endocytosis present	0	3	3
Total	4	7	11

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
