# Peer review of "Endocytosis and Digestion in Carnivorous Pitcher Plants of the Family Sarraceniaceae"

_plants, 2019, doi:10.3390/plants8100367_

Round 1

Reviewer 1 Report

I have several comments that could help the authors to improve the quality of presentation.

Title. First, the words “Update on” looks strange and give the impression that the work simply repeats the known data.  Second, you studied not only the genus Sarracenia but species of two other genera. I would recommend to change the title as follows: Endocytosis and Digestion in Carnivorous Pitcher Plants of the Family Sarraceniaceae.

Abstract, line 25 “Digestive proteins are found in seven species, lacking only in S. flava and in the clade around S. purpurea”. This sentence suggests that a 7+2=9 species were studied. But in lines 19-one could find that eight species of Sarracenia, Heliamphora nutans and Darlingtonia californica (10 species) were analysed. Please, clarify this issue. I recommend mentioning all species names. Digestive proteins are found in species1, species 2…, lacking only in S. flava and in the clade around S. purpurea, where enzyme production is possibly replaced by degradation via the extraordinarydiverse inquiline fauna.  Species 1, species 2, species 3 exhibit both enzyme production and endocytosis; species 4 and species 5 produce enzymes only; no single species shows endocytosis without enzyme production. Abstract should be clear without the main text and table 1.

Abstract, lines 29-30. I suppose that the obtained results are insufficient to draw a conclusion that “endocytotic nutrient uptake obviously evolved from membrane recycling in the frame of the exocytosis of enzymes”. The data obtained show that all (3) species with endocytosis also secreted digestive enzymes, but not vice versa. It is not enough for such evolutionary conclusion concerning general biological phenomenon. Please, reformulate this sentence and keep closer to the actually obtained results.

lines 72-77. The presence of cuticular pores was found in 7 studied Sarraceniaceae species, but in line 19 and table 1 I found that eight species of Sarracenia were tested for cuticular pores. Please, clarify.

lines 177-179 “Growth of trap inhabiting bacteria, which might falsify the outcome by producing additional proteases, was inhibited by adding tetracycline (concentration 0.1 mg per l)”. Why do you think that in this way you managed to suppress the growth of bacteria? Were control experiments performed without the addition of tetracycline? Please, explain.

Author Response

Dear Reviewer 1,

     Thank you very much for your valuable and helpful comments and suggestions. We tried to consider all points in our revised manuscript. These points are also incorporated in the attached pdf-file.

 I have several comments that could help the authors to improve the quality of presentation.

Title. First, the words “Update on” looks strange and give the impression that the work simply repeats the known data.  Second, you studied not only the genus Sarracenia but species of two other genera. I would recommend to change the title as follows: Endocytosis and Digestion in Carnivorous Pitcher Plants of the Family Sarraceniaceae.           

We changed the title following the reviewer’s suggestion.

Abstract, line 25 “Digestive proteins are found in seven species, lacking only in S. flava and in the clade around S. purpurea”. This sentence suggests that a 7+2=9 species were studied. But in lines 19-one could find that eight species of Sarracenia, Heliamphora nutans and Darlingtonia californica (10 species) were analysed. Please, clarify this issue.

  10 species were analyzed; with two subspecies of Sarracenia purpurea resulting in 11 taxa. We changed the confusing text-parts.

I recommend mentioning all species names. Digestive proteins are found in species1, species 2…, lacking only in S. flava and in the clade around S. purpurea, where enzyme production is possibly replaced by degradation via the extraordinary diverse inquiline fauna.  Species 1, species 2, species 3 exhibit both enzyme production and endocytosis; species 4 and species 5 produce enzymes only; no single species shows endocytosis without enzyme production. Abstract should be clear without the main text and table 1.

    Species names were added according to the reviewer´s comments to display all main important results also in the abstract.

Abstract, lines 29-30. I suppose that the obtained results are insufficient to draw a conclusion that “endocytotic nutrient uptake obviously evolved from membrane recycling in the frame of the exocytosis of enzymes”. The data obtained show that all (3) species with endocytosis also secreted digestive enzymes, but not vice versa. It is not enough for such evolutionary conclusion concerning general biological phenomenon. Please, reformulate this sentence and keep closer to the actually obtained results.

   Conclusion in the abstract was changed upon the reviewers’ comments

lines 72-77. The presence of cuticular pores was found in 7 studied Sarraceniaceae species, but in line 19 and table 1 I found that eight species of Sarracenia were tested for cuticular pores. Please, clarify.

   Line 72-77 only refers to the habitus of Sarraceniacea species – Fig 1 does not show all tested species as listed in the abstract (where we talked about the 8 tested Sarracenia species (8 species but 9 taxa with two Sarracenia purpurea ssp.) as listed in table one.

   Line 72 has been changed accordingly for clarification to “The habitus of seven selected species are depicted in Figure 1.”

lines 177-179 “Growth of trap inhabiting bacteria, which might falsify the outcome by producing additional proteases, was inhibited by adding tetracycline (concentration 0.1 mg per l)”. Why do you think that in this way you managed to suppress the growth of bacteria? Were control experiments performed without the addition of tetracycline? Please, explain.

    In earlier experiments nutrient media with tetracycline in the same concentration as used in this study were used to selectively cultivate yeasts and filamentous fungi from the pitcher fluid of S. purpurea. No growth of bacteria was observed on agar plates with tretracycline. Furthermore, resistance against antibiotics are increasingly common but still quite unlikely in the pitchers of carnivorous plants. Therefore, we are convinced that the growth of bacteria in the BSA-Solution used for stimulation was sufficiently reduced.

Reviewer 2 Report

In this article, the authors studied many species of the carnivorous family Sarraceniaceae to examine the proteinase activity and endocytosis, important traits for plant carnivory. This study provides the evidence of endocytotic uptake in Sarraceniaceae where endocytosis was previously undetected due to less extensive taxon sampling, and draw reader’s attention to the previously unappreciated link between enzyme secretion and endocytosis. The methodology is straightforward and the focus of the paper is well phrased. However, there is some room for improvement. One of the major problems is the incomplete sets of provided data. Table 1 summarizes all results, but only small subsets of examined species are shown in each figure. Because readers cannot judge the author’s interpretation, microscopic figures should be provided for all examined species. Another potential problem is an inconsistency with previous work. The protease activity ( https://www.ncbi.nlm.nih.gov/pubmed/9414556 ) and protease secretion ( https://www.ncbi.nlm.nih.gov/pubmed/28812732 ) has been documented in S. purpurea, but the authors failed to appropriately place this work in the context. Because S. purpurea didn’t show protease activity in this study, the authors should discuss in detail about what might have caused the discrepancy. The manuscript can also benefit from the following comments.

Further comments:

Although “enzyme secretion” is consistently discussed, the authors examined protease activity only. No detection of protease activity does not immediately mean no enzyme secretion. I would suggest the authors to explicitly mention this important difference to draw reader’s attention. L29: “obviously”. The evolutionary correlation between the secretion and endocytosis seems to be weak, given the phylogenetic dependency. The occurrence counts summarized in Table 2 makes sense only if evolutionary independence is guaranteed. For example, if 3 species with endocytosis and digestion are monophyletic, most likely the number of trait acquisition is only once. So the author’s interpretation should be described in the modest way, unless the hypothesis was strongly supported after correcting the phylogenetic dependency. Same applies to L139. L77: Fig. 2 should be referenced earlier, maybe in the first sentence of this section. L78: Please define “absorptive zone”. L84: I couldn’t see the “slight” difference in cell size. The authors should provide the area measurement. L152: Nepenthaceaea -> Nepenthaceae It’s difficult to see the permeable cell distribution in Fig. 2a and b because the chunks of cells are framed out. Addition of less-magnified views would increase reader’s understanding. 4: Treatment time should be described as it is discussed but not explained in L120. 5: S. oreaphila -> S. oreophila

Author Response

Dear Reviewer 2,

Thank you very much for your valuable and helpful comments and suggestions. We tried to consider all points in our revised manuscript. These points are also identically listed in the attached pdf-file

In this article, the authors studied many species of the carnivorous family Sarraceniaceae to examine the proteinase activity and endocytosis, important traits for plant carnivory. This study provides the evidence of endocytotic uptake in Sarraceniaceae where endocytosis was previously undetected due to less extensive taxon sampling, and draw reader’s attention to the previously unappreciated link between enzyme secretion and endocytosis. The methodology is straightforward and the focus of the paper is well phrased. However, there is some room for improvement. One of the major problems is the incomplete sets of provided data. Table 1 summarizes all results, but only small subsets of examined species are shown in each figure. Because readers cannot judge the author’s interpretation, microscopic figures should be provided for all examined species.

    Methylene blue: visual microscopical data are provided according to the reviewer`s comments and will be displayed as Supplement Figure 1.

Another potential problem is an inconsistency with previous work. The protease activity ( https://www.ncbi.nlm.nih.gov/pubmed/9414556 ) and protease secretion ( https://www.ncbi.nlm.nih.gov/pubmed/28812732 ) has been documented in S. purpurea, but the authors failed to appropriately place this work in the context. Because S. purpurea didn’t show protease activity in this study, the authors should discuss in detail about what might have caused the discrepancy.

      The first recommended article from Gallie & Chang (1997), (https://www.ncbarticle i.nlm.nih.gov/pubmed/9414556) was added and discussed, the second article from Fukushima et al. 2017 (https://www.ncbi.nlm.nih.gov/pubmed/28812732) is a study on the carnivorous pitcher plant Cephalotus and does not contain information on members of Sarraceniaceae. Instead, we added recent literature Young et al. ( 2018) which we consider more beneficial for this study.

The manuscript can also benefit from the following comments.
Further comments:

Although “enzyme secretion” is consistently discussed, the authors examined protease activity only. No detection of protease activity does not immediately mean no enzyme secretion. I would suggest the authors to explicitly mention this important difference to draw reader’s attention.

    Enzyme secretion was explained as protease production in the whole text.

L29: “obviously”. The evolutionary correlation between the secretion and endocytosis seems to be weak, given the phylogenetic dependency. The occurrence counts summarized in Table 2 makes sense only if evolutionary independence is guaranteed. For example, if 3 species with endocytosis and digestion are monophyletic, most likely the number of trait acquisition is only once. So the author’s interpretation should be described in the modest way, unless the hypothesis was strongly supported after correcting the phylogenetic dependency. Same applies to L139.

    The abstract was changed to “Protease secretion seems to be a prerequisite for endocytotic nutrient uptake.” (Line 29). We removed the evolutionary aspects in the main text. Line 139 was changed.

L77: Fig. 2 should be referenced earlier, maybe in the first sentence of this section.

     We referred to the Methylene blue Figures (Fig. 2 and a new Supplemental Figure 1) earlier, when we first mentioned the methylene blue staining.

L78: Please define “absorptive zone”.

    The term “absorptive zone” was as first used by Lloyd 1942 for the supposed zone for nutrient uptake in pitchers of Sarracenia, Nepenthes etc.; Subsequently this term was used by Juniper et al. (1989) for a zone with the function of absorption, which is obvious for the glandular zone 4 in pitcher plants for Nepenthes (e.g. see page 60). Following their terminology the absorptive zone of the tested plants is comparable to zone 4 in Sarracenia, zone 4 in Heliamphora and zone 5 in Darlingtonia (see Supplement graph 2 on the absorptive zones in all tested genera).

L84: I couldn’t see the “slight” difference in cell size. The authors should provide the area measurement.

    We added a Supplement Figure 3 with the data on cell area measurements. In all measured species of Sarracenia the area of stained cells was larger.

L152: Nepenthaceaea -> Nepenthaceae

    Has been corrected.

It’s difficult to see the permeable cell distribution in Fig. 2a and b because the chunks of cells are framed out. Addition of less-magnified views would increase reader’s understanding.

    The respective images in Fig 2a and 2b have been changed to provide more information.

4: Treatment time should be described as it is discussed but not explained in L120.              

    This information was added in the text.

5: S. oreaphila -> S. oreophila

    Was corrected.

Round 2

Reviewer 1 Report

The authors appropriately revised the manuscript and improved the quality of presentation.

Reviewer 2 Report

> the second article from Fukushima et al. 2017 (https://www.ncbi.nlm.nih.gov/pubmed/28812732) is a study on the carnivorous pitcher plant Cephalotus and does not contain information on members of Sarraceniaceae.

I would suggest the authors to open the full text and run a keyword search with "Sarracenia" and identify their main figure which describes the isolation of secreted enzymes including a protease from Sarracenia purpurea.

Also, "Digestive Enzymes" in Table 1 could be replaced with "Protease activity" so that it does not imply the absence of enzymes other than proteases in non-detected species.